# Learning Human-Perceived Fakeness in AI-Generated Videos via Multimodal LLMs

## Abstract

Can humans identify AI-generated (fake) videos and provide grounded reasons? While video generation models have advanced rapidly, a critical dimension – whether humans can detect *deepfake traces* within a generated video, *i.e.*, spatiotemporal grounded visual artifacts that reveal a video as machine generated – has been largely overlooked. We introduce DEEPTRACEREWARD, the first fine-grained, spatially- and temporally-aware benchmark that annotates human-perceived fake traces for video generation reward. The dataset comprises 4.3K detailed annotations across 3.3K high-quality generated videos. Each annotation provides a natural-language explanation, pinpoints a bounding-box region containing the perceived trace, and marks precise onset and offset timestamps. We consolidate these annotations into 9 major categories of deepfake traces that lead humans to identify a video as AI-generated, and train multimodal language models (LMs) as reward models to mimic human judgments and localizations. On DEEPTRACEREWARD, our 7B reward model outperforms GPT-5 by 34.7% on average across fake clue identification, grounding, and explanation. Interestingly, we observe a consistent difficulty gradient: binary fake *v.s.* real classification is substantially easier than fine-grained deepfake trace detection; within the latter, performance degrades from natural language explanations (easiest), to spatial grounding, to temporal labeling (hardest). By foregrounding human-perceived deepfake traces, DEEPTRACEREWARD provides a rigorous testbed and training signal for socially aware and trustworthy video generation.

## 1 Introduction

Recent advances in video generation technologies, including Veo3 (Anil et al., 2024), Sora (OpenAI, 2024), Pika (Pika, 2024), Meta Movie Gen (AI, 2024), Gen-3 (Runway Research, 2024), Kling (Kling, 2024), and others (Yang et al., 2024; Team, 2024; MiniMax, 2024; Wang et al., 2023a; Li et al., 2024b), have demonstrated remarkable capabilities in producing increasingly realistic videos. Alongside this progress, numerous studies about video generation have been conducted (Huang et al., 2023; Liu et al., 2023; Bansal et al., 2024; Huang et al., 2024; Liu et al., 2025), such as evaluating video prompt alignment towards a set of provided prompts as in VBench (Huang et al., 2023), or analyzing the physical commonsense in deepfake videos as explored by VideoPhy (Bansal et al., 2024), *etc*. However, these evaluations primarily compare AI-generated videos against a set of predetermined criteria, neglecting one of the most fundamental aspect:

> Can humans distinguish AI-generated videos from natural videos and provide grounded reasons for their judgments?

This paper aims to emphasize the critical aspect of human visual perception on AI-generated videos, as more responsible and trustworthy AI is needed (Harris, 2021; Twomey et al., 2023). We argue that human-perceived "deepfake traces" – grounded visual artifacts and inconsistencies that reveal machine generation – are essential for video generation models. We introduce DEEPTRACEREWARD, the first benchmark of human-perceived deepfake traces with fine-grained, spatiotemporally grounded expert annotations. As illustrated in Figure 2, we collect high-quality, realistic-style videos from seven state-of-the-art (SOTA) video generators and provide expert-level, fine-grained annotations through the LabelBox (LabelBox, 2024) interface. The dataset comprises 3.3k generated videos with

4.3k detailed annotations and 3.3k real videos for experiment purposes. Each annotation (i) provides a natural-language explanation, (ii) localizes the perceived deepfake trace with bounding boxes across frames, and (iii) marks precise onset and offset timestamps. Despite strong surface realism, we find that generated videos often betray their artificial nature through movement-related anomalies, ranging from low-level visual artifacts such as object distortion, to higher-level commonsense violations like the unnatural disappearance of objects. Inspired by these findings, we systematically analyze and categorize annotated deepfake traces into nine major categories, as shown in Figures 1, 4 and 6.

To benchmark performance, we conduct extensive experiments with 13 baseline multimodal language models (LMs), evaluating their capability to capture human visually perceived deepfake traces within videos on DEEPTRACEREWARD. Interestingly, we find that although several SOTA multimodal LMs – including GPT 5 (Achiam et al., 2023) and Gemini 2.5 Pro (Team et al., 2023) – achieve high accuracy (>70%) on binary real *vs* fake video classification, their ability to accurately ground fine-grained deepfake traces remains limited, with performances only ranging below 36%.

We further conducted experiments to train an improved reward model using our collected DEEPTRAC-EREWARD dataset. Building upon Video LLaMa 3 (Zhang et al., 2025), our 7B model achieves an average performance of 70.2% across identification, grounding, and explanation of deepfake traces, surpassing GPT 5 and Gemini 2.5 Pro by 34.7% and 40.2%, respectively. Notably, we observe a clear trend: binary fake *vs* real video classification is consistently easier for reward modeling than the more challenging task of deepfake trace detection – our trained model can reach 99.4% for the classification task but  70% for others. Moreover, within the latter, the difficulty increases progressively – from natural language explanations (easiest), to bounding boxes, to temporal labeling (hardest). We believe DEEPTRACEREWARD will serve as a valuable resource for collecting and analyzing fine-grained human perceived fakeness on AI-generated videos.

## 2 DEEPTRACEREWARD DATASET

Our goal is to collect fine-grained, high-quality annotations for human-perceived deepfake traces in AI-generated videos. We aim to investigate what kinds of fake cues humans can identify while watching these videos, and to explore the detection gap between human perception and machine predictions. This, in turn, offers deeper insights into the challenges and future directions for achieving more robust video understanding and generation. In this section, we present our two-stage data curation pipeline (§2.1), which includes prompt design and video collection. We then describe the annotation process of DEEPTRACEREWARD using the LabelBox interface (LabelBox, 2024) (§2.2), followed by an analysis of the dataset's key features and statistics (§2.3).

### 2.1 VIDEO COLLECTION

To curate our dataset, we first use GPT 4 (Achiam et al., 2023) to generate natural and realistic prompts. These prompts are manually filtered and then fed into various text-to-video (T2V) models to synthesize candidate videos. A subsequent manual filtering step retains only high-quality, realistic videos. During this stage, we discard samples exhibiting severe visual degradation, implausible physical interactions, or incoherent motion throughout the whole video.

**What kind of videos should we collect?** Two key criteria in our video collection process is to include only **high-quality** generated videos that **contain motion**. The first criterion is motivated by annotation challenges observed in low-quality videos generated by many open-source models, which tend to be ambiguous, extremely short (e.g., only 1 second), or entirely distorted across all frames – making them unsuitable for fine-grained deepfake trace identification. The second criterion comes from our initial observations, that *humans sometimes cannot tell an AI-generated video as fake*, especially if the video is a still one. We apply manual filtering on collected videos to preserve the ones that depict dynamic scenes involving object or human movement – artifact patterns such as unnatural trajectories, object distortions, and sudden blurring are far more likely to emerge in movement-rich scenarios than in static scenarios, which rarely exhibit consistent visual anomalies. Even after the manual filtering, throughout the annotation process, in 6.0% videos annotators find they can't tell if it's AI or not.

We collect generated videos directly using the following models: Kling 1.0 and Kling 1.5 (Kling, 2024), Pika 1.5 (Pika, 2024), and Mochi 1 (Team, 2024). For OpenAI's Sora (OpenAI, 2024),

Object Disappearance. *Explanation:* Two men in dark clothes suddenly disappeared while walking to the stairs.

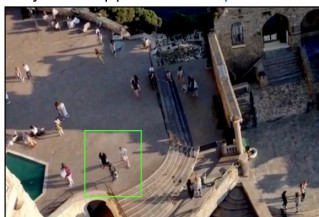 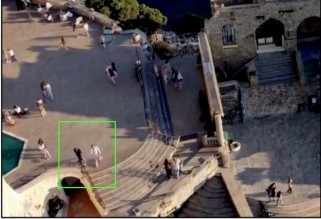 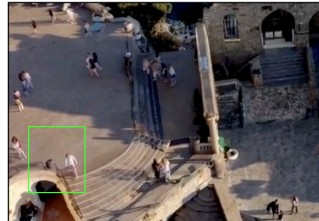

Object Distortion. *Explanation:* The kettle in the man's hand is undergoing a fluid-like deformation.

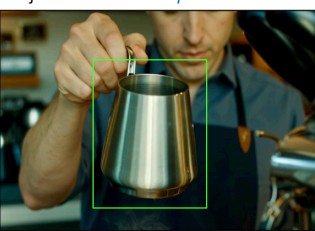 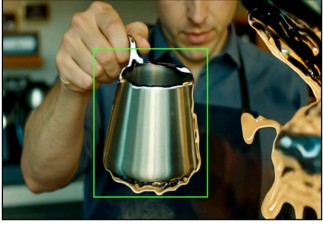 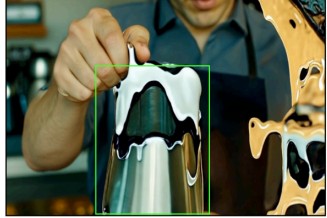

Object Splitting. *Explanation:* As the goalkeeper moved to defend, his body unnaturally split into two separate figures.

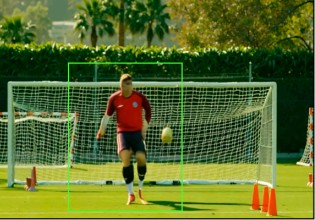 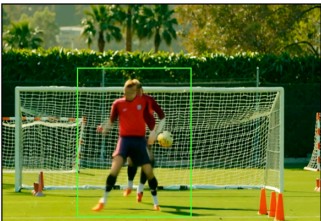 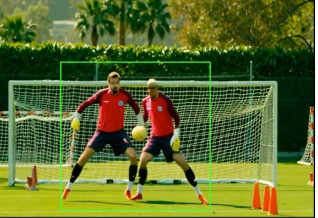

Object Merging. *Explanation:* Two otters merge into one while moving.

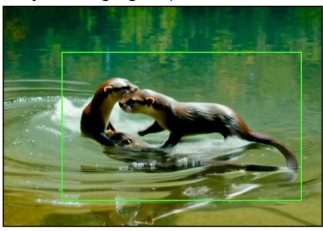 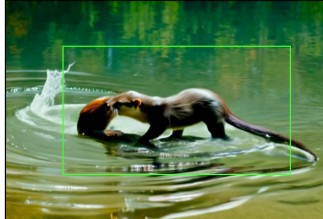 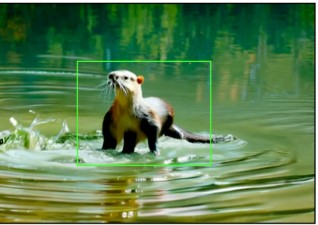

Figure 1: **Human-perceived deepfake traces examples.** The shown cases are selected from *Pika 1.5*, *MiniMax-Video-01*, and *Sora* generated videos. For each deepfake trace, we annotate local bounding box regions, start and end timestamps, and provide natural language explanation. All fake trace categories are summarized in Section 2.3 and distribution can be found in Figure 4.

we manually curated demonstration videos available on its official website.[1] For MiniMax-Video-01 (MiniMax, 2024) and Gen-3 (Runway Research, 2024), we selected high-quality samples from generations released by VBench (Huang et al., 2023). In total, we collected 3,318 unique high-quality fake videos. To support the downstream goal of teaching multimodal language models to distinguish deepfake traces, we also include real videos for training purposes. We sample an equal number (3,318) of real videos from the high-quality LLaVa-Video-178K (Zhang et al., 2024) dataset. These videos are clipped to match the length distribution of the fake videos, ensuring that for each video length, the number of real and fake videos is balanced.

---

[1]https://openai.com/sora/

Figure 2: **DEEPTRACEREWARD data curation pipeline.** Selected videos are uploaded to our annotation platform LabelBox (LabelBox, 2024), where experts provide fine-grained deepfake trace annotations with bounding boxes, textual explanations, and start / end timestamps.

## 2.2 ANNOTATION PIPELINE

The filtered set of 3,318 high-quality AI-generated fake videos is subsequently annotated by expert annotators using the LabelBox (LabelBox, 2024) platform, as illustrated in Figure 3. Annotators conduct meticulous frame-by-frame inspections, labeling each video with temporally-aware bounding boxes that spatially localize regions exhibiting visual anomalies. Each annotation is further enriched with structured category tags reflecting the type of deepfake trace (e.g., distortion, blurring, merging, etc.), as defined in Section 2.3. In addition to spatial and categorical annotations, annotators are instructed to provide natural language explanations that describe the context and nature of each fake clue. These explanations are critical for enabling fine-grained supervision in downstream model training and evaluation. Due to the time-intensive nature of this task, detailed explanations are provided for 62.7% of the annotated deepfake traces. In total, this annotation effort results in 4,334 unique expert-labeled deepfake traces across 3,318 AI-generated videos.

| Video Source | # Data | % Explanation | Avg. Resolution | Video Length (s) | Trace Length (s) |
|---|---|---|---|---|---|
| | | *Fake Videos* | | | |
| Kling 1.0 (Kling, 2024) | 1,264 | 58.4% | $720 \times 1280$ | 5.1 | 3.5 |
| Sora (OpenAI, 2024) | 38 | 58.2% | $853 \times 1433$ | 16.3 | 7.3 |
| Pika 1.5 (Pika, 2024) | 2,215 | 65.4% | $720 \times 1296$ | 5.0 | 3.7 |
| Kling 1.5 (Kling, 2024) | 226 | 59.3% | $1080 \times 1920$ | 5.1 | 4.1 |
| MiniMax-Video-01 (MiniMax, 2024) | 78 | 74.4% | $720 \times 1280$ | 5.6 | 4.9 |
| Mochi 1 (Team, 2024) | 102 | 67.6% | $480 \times 848$ | 5.4 | 3.8 |
| Gen-3 (Runway Research, 2024) | 411 | 59.9% | $768 \times 1280$ | 10.7 | 6.6 |
| Overall | 4,334 | 62.7% | $739 \times 1313$ | 5.7 | 4.0 |
| | | *Real Videos* | | | |
| LLaVA-Video-178K (Zhang et al., 2024) | 3,318 | - | $623 \times 1055$ | 5.78 | - |

Table 1: DEEPTRACEREWARD benchmark statistics. Video and trace lengths are represented in seconds (s) as average values. The fake video collection includes 7 diverse state-of-the-art (sota) model sources, while real videos are randomly sampled from the LLaVA-Video-178K (Zhang et al., 2024) dataset, clipped to same video length distributions of the fake videos.

**Annotation Challenges.** Despite our structured pipeline, the annotation process poses several challenges. A primary issue was *subjective ambiguity* – annotators occasionally struggled to distinguish between closely related artifact types (e.g., object merging vs. object disappearance). To mitigate this problem, we adopt a consensus-based workflow in which annotators collaboratively look at the same ambiguous case and agree with the majority-vote results. Each annotated video is then reviewed by one to two additional annotators to ensure cross-validation and minimize bias. Another challenge involved linguistic inconsistency in the natural language explanations. While some annotators use concise, objective descriptions, others may provide more interpretive commentary. To improve consistency across the dataset, we deploy GPT-4 to post-process and standardize the explanation text, ensuring a more uniform and model-friendly annotated corpus.

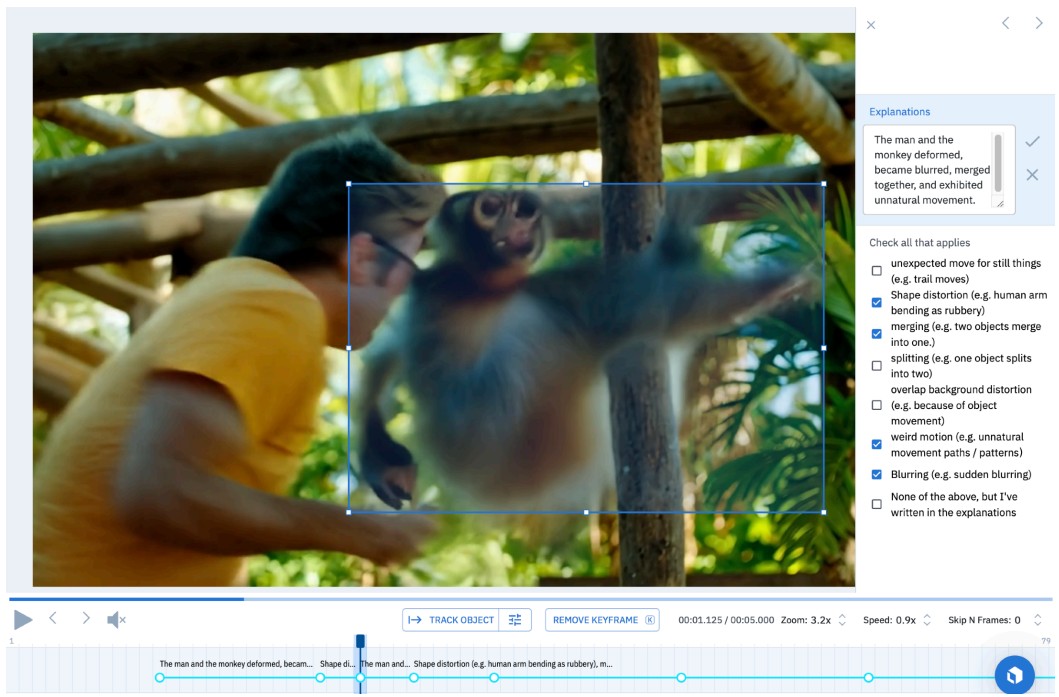

Figure 3: **Labelbox annotation interface.** Each video is annotated with localized bounding boxes that highlight specific regions across frames where fakeness is perceived. Each annotated deepfake trace is accompanied by a natural language explanation and predefined category labels.

## 2.3 DATASET ANALYSIS

**Statistics.** Table 1 summarizes the composition of the DEEPTRACEREWARD benchmark. It includes 4,334 deepfake trace annotations on 3,318 fake videos sourced from six sota T2V models, and paired with 3,318 real videos sourced from LLaVA-Video-178K (Zhang et al., 2024) that are clipped to same video length distributions as in the fake videos for subsequent training purposes. For each source, we report the number of videos, proportion of the ones with human-written explanations, average resolution (computed as the mean of height and width separately), average video length in seconds, and the average fake clues length as we annotated the start and end timestamps. The dataset captures significant variation in resolution and temporal length across models.

**Deepfake Trace Category.** We further analyze the annotated deepfake trace reasons, and summarize them into 9 major movement-centric categories. Due to the time-intensive nature of this task, annotators provide detailed categories for 60.9% of the annotations. Within which, we summarize 9 major reasons that take up 90% cases, with the remaining 10% covering multiple minor categories such as light effect, liquid motion, shadowing, ..., etc. Notice that one annotation can fall into multiple fake reason categories, *e.g.*, one deepfake trace can both include object blurring and unexpected move. Therefore, we show the relative distribution (frequency) of these nine artifact categories as illustrated in Figure 4. The category definitions are as follows, and concrete examples for each category are demonstrated in Figures 1 and 6.

**Object Distortion:** This type refers to cases where objects exhibit abnormal shape distortions, such as a kettle appearing to melt, or arms bending like rubber.

**Sudden Blurring:** This type refers to abrupt visual degradation, such as a puppy suddenly becoming blurred, or a face losing definition mid-conversation.

**Object Trajectory:** This type refers to objects moving in unnatural paths, such as a train barrier sliding forward incorrectly, or a ball sharply curving mid-air without cause.

**Redundant Object:** This type refers to the appearance of extraneous elements, such as a third arm appearing during a gesture, or an extra tree emerging in the background as someone runs.

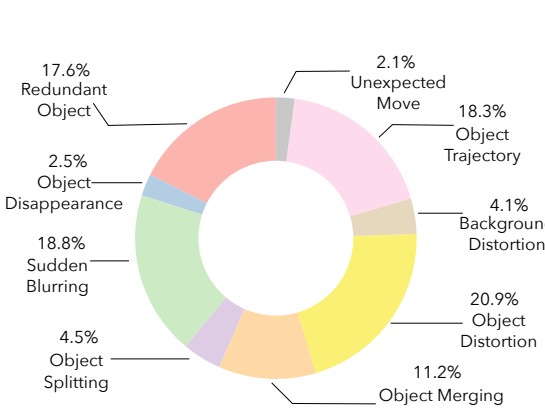

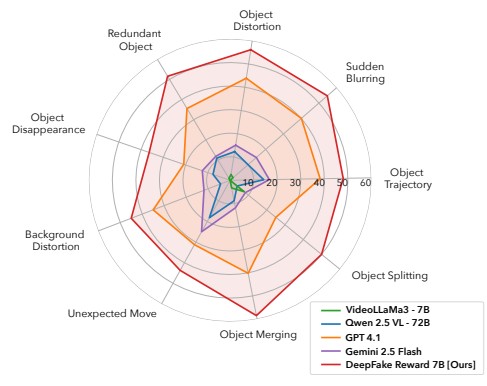

Figure 4: DEEPTRACEREWARD deepfake trace category statistics. Category definitions can be found in Section 3.3, and concrete examples for each category are listed in Figures 1 and 6.

Figure 5: Performance analysis between baseline models and our best reward model trained on the collected DEEPTRACERE-WARD dataset. Our model is much better in all categories, especially in "object spltting" and "object merging".

**Object Merging:** This type refers to cases where distinct objects fusing together, such as two otters blending into a single shape, or two dancers becoming visually indistinguishable.

**Object Splitting:** This type refers to a single object dividing into multiple parts, such as a goalkeeper's body splitting into two mid-motion.

**Background Distortion:** This type refers to unrealistic warping or deformation of the background, such as a parked car stretching as someone walks by, or rippling walls.

**Object Disappearance:** This type refers to sudden vanishing of visible elements, such as a person disappearing mid-step, or a soccer ball vanishing mid-kick.

**Unexpected Move:** This type refers to inexplicable motion of typically static objects, such as a beer glass sliding on its own, or a stationary chair shifting position.

## 3 EXPERIMENTS

In this section, we aim to address the following questions: Do current multimodal language models (LMs) possess human-level visual intelligence to identify human-perceived deepfake traces? If not, can we teach them to do so using DEEPTRACEREWARD? We begin by describing our experimental setup and baseline models (§3.1). While humans can reliably identify deepfake traces, we find that DEEPTRACEREWARD presents significant challenges for existing models. We then detail our supervised fine-tuning (SFT) experiments using two base models: VideoLLaMA 3 (Zhang et al., 2025) and Qwen 2.5 VL (Bai et al., 2025) (§3.2). Finally, we provide a comprehensive analysis of both baseline and trained model results (§3.3). This includes in-depth comparison between baseline models and our models, impact of different supervision, and an error analysis.

### 3.1 EXPERIMENTAL SETUPS

**Multimodal Language Models** We evaluate DEEPTRACEREWARD on 13 recent multimodal LLMs, including GPT 5 and GPT 4.1 (Achiam et al., 2023), Gemini 2.5 Pro and Gemini 2.5 Flash (Anil et al., 2024), Video-LLaVa 7B (Lin et al., 2023), LLaVa-One-Vision 7B (Li et al., 2024a), Phi-3.5-Vision (Microsoft, 2024), Phi-4-Vision (Abdin et al., 2024), Qwen 2 VL 7B (Bai et al., 2023), Qwen 2.5 VL 7B, 32B, 72B (Bai et al., 2025) and VideoLLaMA3 7B (Zhang et al., 2025). We employ VLMEvalKit (Duan et al., 2024) to rigorously evaluate multimodal language models and ensure reproducibility. To facilitate fair comparisons, we maintain consistent prompts and configurations

| Model | Fake *v.s.* Real Classification | | | | Deepfake Trace Detection | | | |
|---|---|---|---|---|---|---|---|---|
| | Overall ↑ | Acc. ↑ | Fake Acc. ↑ | Real Acc. ↑ | Explan. ↑ | BBox IoU ↑ | BBox Dist. ↓ | Time Dist. ↓ |
| *Baseline Models* | | | | | | | | |
| GPT 5 (Achiam et al., 2023) | 35.5 | 90.7 | 84.6 | 98.8 | 40.9 | 10.4 | 37.0 | 100.0 |
| GPT 4.1 (Achiam et al., 2023) | 36.5 | 92.9 | 89.1 | 97.9 | 31.3 | 21.8 | 23.7 | 100.0 |
| Gemini 2.5 Pro (Team et al., 2023) | 30.0 | 84.3 | 75.7 | 95.8 | 26.8 | 8.9 | 44.7 | 100.0 |
| Gemini 2.5 Flash (Team et al., 2023) | 23.0 | 74.5 | 56.8 | 97.9 | 10.3 | 6.3 | 68.7 | 99.0 |
| LLaVa-One-Vision 7B (Li et al., 2024a) | 11.6 | 46.4 | 38.4 | 56.9 | 0.0 | 0.0 | 98.2 | 100.0 |
| Video-LLaVa 7B (Lin et al., 2023) | 10.8 | 43.0 | 0.0 | 100.0 | 0.0 | 0.0 | 100.0 | 100.0 |
| Phi-4-vision (Abdin et al., 2024) | 8.9 | 35.5 | 3.2 | 78.3 | 0.1 | 0.0 | 99.0 | 100.0 |
| Phi-3.5-Vision (Microsoft, 2024) | 6.5 | 25.8 | 7.7 | 49.7 | 0.2 | 0.0 | 98.4 | 100.0 |
| Qwen 2 VL 7B (Bai et al., 2023) | 15.0 | 56.7 | 28.6 | 94.0 | 3.1 | 0.0 | 86.7 | 100.0 |
| Qwen 2.5 VL 7B (Bai et al., 2025) | 15.7 | 51.7 | 20.2 | 93.4 | 10.5 | 0.6 | 87.9 | 99.9 |
| Qwen 2.5 VL 32B (Bai et al., 2025) | 13.5 | 47.4 | 8.9 | 98.5 | 5.1 | 0.0 | 95.5 | 98.4 |
| Qwen 2.5 VL 72B (Bai et al., 2025) | 17.3 | 50.0 | 16.6 | 94.3 | 7.4 | 0.1 | 90.7 | 88.2 |
| VideoLLaMa3 7B (Zhang et al., 2025) | 10.0 | 38.1 | 4.3 | 82.8 | 1.8 | 0.0 | 100.0 | 100.0 |
| DEEPTRACEREWARD *Models* | | | | | | | | |
| Our (base Qwen 2.5 VL 7B) | 38.4 | 74.7 | 55.7 | 100.0 | 33.3 | 1.7 | 63.0 | 56.2 |
| w/o time | 29.6 | 79.8 | 64.6 | 100.0 | 37.2 | 1.2 | 57.7 | 100.0 |
| w/o explanation | 40.0 | 91.3 | 85.0 | 99.7 | 0.0 | 1.8 | 46.0 | 33.3 |
| w/o time & w/o explanation | 18.3 | 72.3 | 51.4 | 100.0 | 0.0 | 1.1 | 66.8 | 100.0 |
| **Our (base VideoLLaMa3 7B)** | **70.2** | 99.4 | 99.3 | 99.4 | 70.6 | **32.6** | 13.6 | 21.9 |
| w/o time | 50.8 | 99.1 | 98.9 | 99.4 | **71.6** | 32.4 | 14.0 | 100.0 |
| w/o explanation | 52.4 | 99.2 | **99.6** | 98.8 | 0.0 | 32.0 | **13.5** | **21.6** |
| w/o time & w/o explanation | 32.8 | **99.6** | **99.6** | **99.7** | 0.0 | 31.5 | 13.7 | 100.0 |

Table 2: Test set results on DEEPTRACEREWARD. All baseline models achieve below 37% performance regardless of their sizes. The sota models GPT 5, GPT 4.1, and Gemini 2.5 Pro are the only ones to have an overall score over 30%. In contrast, our best 7B model based on VideoLLaMa 3 can easily surpass GPT 5 by 34.7%, and Gemini 2.5 Pro by 40.2%, reaching 70.2% after training on our high-quality DEEPTRACEREWARD dataset. Interesting, we can observe a consistent difficulty gradient: binary classification is substantially easier than fine-grained deepfake trace detection; within the latter, performance degrades from natural language explanations (easiest), to spatial grounding, to temporal labeling (hardest). ↑ means higher is better, ↓ means lower is better.

across all models, whenever permitted by the model specifications. Detailed information regarding the prompt and experimental settings is provided in Appendix §E.

**Evaluation Metrics**  We evaluate deepfake trace detection using a comprehensive set of seven metrics, with ↑ meaning a higher score is and ↓ meaning a lower score is better: (1) **Accuracy (↑)** is the classification performance over all of the fake and real videos. (2) **Fake Accuracy (↑)** is the classification performance over all of the fake videos, included for analysis purposes since some models tend to always predict REAL. (3) **Real Accuracy (↑)** is the classification performance over all of the real videos, included for analysis purposes since some models tend to always predict FAKE. (4) **Explanation (↑)** score refers to the GPT 4.1 judgment score for the explanations generated. Specifically, we ask GPT 4.1 to rank the generated explanation to 0, 0.5, or 1, representing total incorrectness, partial correctness, and total correctness, comparing to the ground-truth explanation. We skip the instances where either the ground-truth explanation is unannotated or the ground-truth is a real video. Detailed evaluation prompt is in Appendix D. (5) **BBox IoU (↑)** is the Intersection over Union (IoU) that evaluates the quality of deepfake trace region bounding-box generation, defined as

$$\text{IoU} = \frac{|A_{\text{pred}} \cap A_{\text{gt}}|}{|A_{\text{pred}} \cup A_{\text{gt}}|},$$

where $A_{\text{pred}}$ and $A_{\text{gt}}$ denote the areas of predicted and ground-truth bounding boxes, respectively. We convert the bounding box coordinate values into ratios for scale invariance. (6) **BBox Distance (↓)** is defined as the Euclidean distance between the center points of the predicted deepfake trace bounding box and the ground-truth annotation. To ensure scale invariance, bounding box coordinates are first converted into ratios, and the resulting distance is normalized by $\sqrt{2}$. (7) **Time Distance(↓)** is defined as the distance between the predicted starting second of the deepfake trace and the ground-truth annotation. Seconds are converted into ratios over the whole video lengths.

**Overall (↑)** score is the combined evaluation considering most of above:

$$\text{Overall} = \frac{\text{Accuracy} + \text{Explanation\_score} + \text{BBox\_IoU} + (100 - \text{Time\_distance})}{4}.$$

## 3.2 TRAINING SETUPS

We apply supervised-finetuning (SFT) on two different state-of-the-art video understanding base models: VideoLLaMA 3[2] (Zhang et al., 2025) and Qwen 2.5 VL[3] (Bai et al., 2025). Train, val, test sets are randomly split as 8:1:1 by unique videos in the DEEPTRACEREWARD dataset, with details in Table 3. Details about hyperparameters and training setups can be found in Appendix G. The default question prompt is "`<video>` Decide whether the video is AI-generated or real by detecting unnatural parts. If you don't detect any unnatural parts and think the video is real, reply with REAL. Otherwise, if you detect any, reply with FAKE, and provide the coordinates of the unnatural parts in [x0, y0, x1, y1] format, the starting time of them, and an explanation.". Then, the default answer prompt follows "FAKE. The video is AI-generated. The unnatural part is at [BBox] starting [Time] seconds. The reason is because [Explanation]" or "REAL. The video is real. There is no unnatural part.", where [BBox] and [Time] use absolute values. We include three types of additional settings for analysis comparisons: one without using the temporal annotation, one without using the annotated textual explanations, and one without either of them.

## 3.3 RESULTS AND ANALYSIS

We highlight several key observations and analyses from the test set experiment results in Table 2.

**Baseline models perform poorly regardless of their sizes.** All baseline models achieve below 37% on overall performance, with the sota models GPT 5, GPT 4.1, and Gemini 2.5 Pro being the only ones to exceed 30%, while Gemini 2.5 Flash only reaches 23%. Interestingly, GPT 4.1 is better than GPT 5 by 1% on the overall score, with GPT 5 producing stronger explanations, and GPT 4.1 localizing deepfake traces' local regions more accurately (higher BBox IoU and lower BBox distance). Looking at the results of different sizes of the Qwen 2.5 VL models, we see that scaling within the family is not monotonic (7B model better than 32B model on overall score). Comparing the baselines' performance on binary classification and deepfake trace detection, we can easily find that they generally have a higher score on the former task. All models also consistently show a strong "REAL" bias; for instance, Qwen 2.5 VL 32B reaches 98.5% classification accuracy on real videos but 8.9% accuracy on fake (AI-generated) ones. Among all the evaluation metrics, we can see that temporal prediction, with metric being time distance (↓), is the hardest criterion for all models: all baselines except Qwen 2.5 VL 72B have time distance close to 100 (out of 100).

**Our best 7B model surpasses sota models GPT 5 and Gemini 2.5 Pro by large margins under all metrics.** In contrast, our best-performing 7B model based on VideoLLaMa 3 demonstrates substantial performance improvements in fake real video classification as well as deepfake trace identification. It can easily surpass GPT 5 by +34.7%, GPT 4.1 by +33.7%, and Gemini 2.5 Pro by +40.2% on overall score, reaching 70.2% after training on our high-quality dataset. Looking closely on the individual metrics, our model can always reach 99%+ accuracy on binary classification, and 70.6% on explanation performance (under LLM as a judge). It also reaches 32.6 (out of 100) on bounding box IoU ↑ evaluation, and 13.6 (out of 100) ↓ on bounding box distance. Notably, with all baseline models stuck on the temporal prediction – reaching almost 100 (out of 100) on time distance(↓) – our best model achieves 21.6 (out of 100) on time distance.

**Consistent difficulty gradient.** The consistent pattern holds for both baselines and our models: binary real *v.s.* fake video classification is substantially easier than the fine-grained deepfake trace detection task; within the latter, performance degrades from natural-language explanations (easiest), to spatial grounding, to temporal localization (hardest). Sota baseline models such as GPT and Gemini can in average achieve 85.6% accuracy on classification task, but often stuck on the fine-grained detection tasks, reaching in average 27.3 (out of 100) score on explanation (↑), 11.9 (out of 100) on localization bounding box IoU (↑), and 99.8 (out of 100) score on time distance (↓). As for our

---

[2] https://github.com/DAMO-NLP-SG/VideoLLaMA3
[3] https://github.com/QwenLM/Qwen2.5-VL

best 7B model, it achieves 99.4% accuracy on classification task, but for the fine-grained detection tasks, it reaches 70.6 (out of 100) score on explanation ($\uparrow$), 32.6 (out of 100) on localization bounding box IoU ($\uparrow$), and 21.9 (out of 100) score on time distance ($\downarrow$). While largely surpassing all baseline models, these numbers are still far from perfect as humans would do.

**Ablation studies on supervision signal controls.** We conduct ablation studies to investigate whether training the model to output explanations versus spatiotemporal groundings interferes with each other, since the former outputs natural language and the latter outputs numbers. To this end, we control the supervision signals that we feed into our model during training. We include three types of settings for the supervision control: *[w/o time]*, where temporal annotations (i.e., when the deepfake trace starts in seconds) are removed; *[w/o explanation]*, where natural language explanations for the deepfake traces are removed; and *[w/o time & w/o explanation]*, where both are removed. As shown in Table 2, we can see that the setting *[w/o explanation]* indeed achieves the best performance score on the bounding box distance metric and time distance metric. Moreover, under the *[w/o time & w/o explanation]* setting, our trained model achieves highest classification scores of 99.6%. Overall, the fine-grained metric scores show only minor differences across settings, and incorporating all supervision signals during training yields the best overall performance.

**Error analysis comparison.** We further conduct a qualitative error analysis by comparing generations from our model with those of the strongest baseline, GPT 4.1. Specifically, regarding bounding box Intersection over Union (IoU), we observe that GPT 4.1 frequently (in approximately 64% of all cases) defaults to predict the entire video frame (e.g., bounding box coordinates "[0, 0, 1280, 720]" for a 720 * 1280 resolution video) regardless of the actual content. In contrast, our best 7B model consistently localizes deepfake traces more accurately. Furthermore, as illustrated in Figure 7, our best 7B model provides more precise and detailed explanations by not only accurately identifying distorted objects but also articulating the specific nature of their distortions, surpassing GPT-4.1 in both grounding and interpretability.

## 4 RELATED WORK

**Video Generation and Evaluations** Text-to-Video (T2V) generation produces videos from textual prompts using Transformer and diffusion models (Vaswani et al., 2023; Ho et al., 2020). Closed-source systems (OpenAI, 2024; AI, 2024; Pika, 2024; Runway Research, 2024; Kling, 2024) showcase strong visual quality, while open-source models like Mochi and CogVideoX also achieve competitive results (Peebles & Xie, 2023; Team, 2024; Yang et al., 2024). Evaluation typically relies on vision-based scores such as IS, FID, and FVD, or multimodal benchmarks like VBench and VideoPhy (Barratt & Sharma, 2018; Heusel et al., 2018; Unterthiner et al., 2019; Huang et al., 2023; Bansal et al., 2024). However, these methods emphasize predefined attributes (e.g., object count, appearance, alignment) and overlook the human-centric question of whether viewers can identify concrete fake cues. While VBench and human-preference studies (Liu et al., 2025) provide holistic scores, they lack fine-grained localization. Our work instead gathers spatially and temporally grounded human annotations of perceived fakeness, offering a precise view of how humans judge generated videos.

**AI-generated Video Detection** Prior research has extensively explored the detection of machine-generated text (Dugan et al., 2024; Ippolito et al., 2020) and AI-generated images (Guo et al., 2023; Lorenz et al., 2023; Wu et al., 2023; Wang et al., 2023b). With the emergence of video generation models, several techniques have also been proposed for fake video detection (Gu et al., 2022; Xu et al., 2024; Gu et al., 2021). However, most of these efforts are narrowly focused on human face deepfakes, whereas current video generation models are capable of producing much more diverse and complex content. Recent work such as Beyond the Imitation Game (Vahdati et al., 2024) attempts to detect synthetic videos by learning generalized traces of video generation artifacts. Similarly, DeMamba (Chen et al., 2024) introduces the large-scale GenVideo dataset, comprising both fake and real videos, and improves detection performance by leveraging spatiotemporal inconsistencies. Despite these advancements, both methods treat fake video detection as a binary classification problem and do not require models to explain or localize the underlying reasons for fakeness.

## 5 CONCLUSION

In this paper, we introduce DEEPTRACEREWARD, the first large-scale benchmark with expert-annotated, spatially and temporally localized deepfake traces. We show that existing multimodal LMs fall short in deepfake trace detection. By training a dedicated reward model on DEEPTRACEREWARD, we demonstrate significant performance gains. We hope DEEPTRACEREWARD will drive future research toward more human-aligned video generation and understanding.

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

## A  ADDITIONAL EXAMPLES

Sudden Blurring. *Explanation:* As the puppy plays among the leaves, it becomes blurred along with the leaves as it moves.

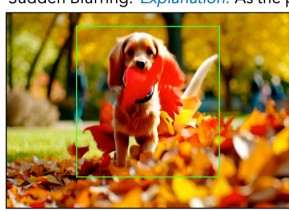 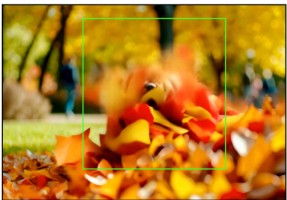 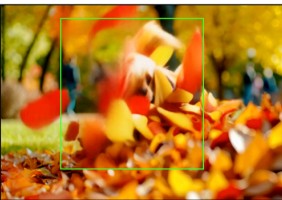

Object Trajectory. *Explanation:* While the camera was still, the red-and-white train barrier moved forward on its own.

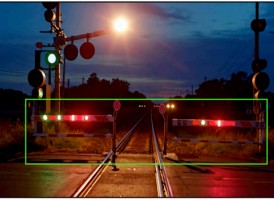 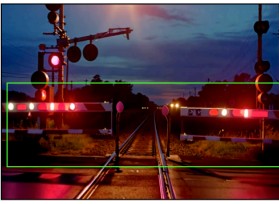 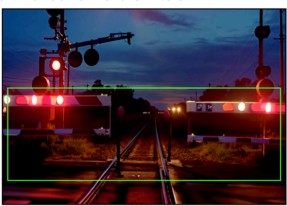

Redundant Object. *Explanation:* As the man forms a heart shape with his hands, a third arm suddenly appears.

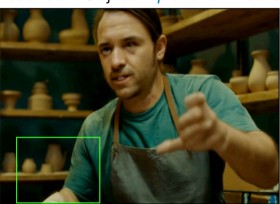 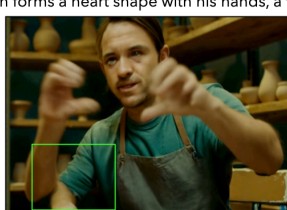 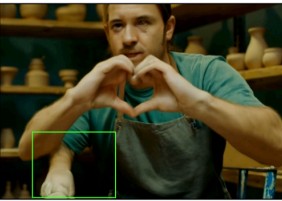

Unexpected Move. *Explanation:* The beer glass unnaturally tilts and moves toward the person's mouth without any visible external force.

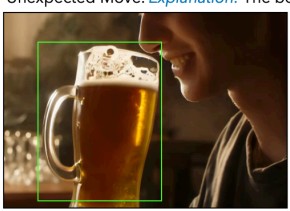 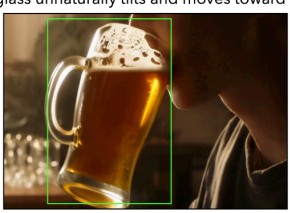 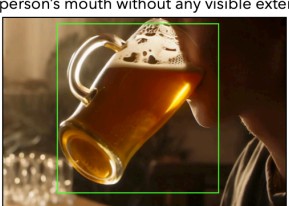

Background Distortion. *Explanation:* As the pedestrian crosses the street, the car in the background undergoes irregular deformation.

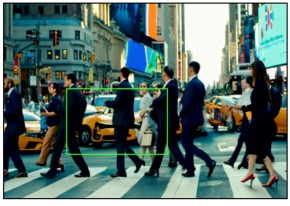 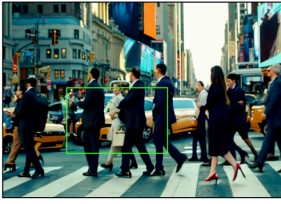 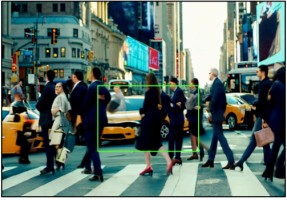

Figure 6: **DEEPTRACEREWARD examples by category.** Category definitions are in Section 2.3 and dataset statistics can be found in Figure 4.

## B  ETHICS STATEMENT

Our work proposes DEEPTRACEREWARD, a reward dataset aiming at advancing academic research. We have manually filtered our dataset multiple times to ensure that there is no unsafe content in it. In a broader perspective, DEEPTRACEREWARD proposes a new way that humans can provide visual perception feedback to AI-generated videos, and makes fake videos more interpretable by tracing

deepfake traces. On the other hand, if misused, the dataset may be used to train video generators to produce higher quality fake videos that could be used for deceptive purposes.

## C    LIMITATIONS AND FUTURE DIRECTIONS.

First, DEEPTRACEREWARD requires intensive human effort in annotation during the whole collection process. Annotators can make minor mistakes during this process. Second, this work focuses on existing off-the-shelf video generators and multimodal LMs. Future work may explore the training effect of DEEPTRACEREWARD on video generation tasks. For example, we can use DEEPTRAC-EREWARD as a fine-grained reward model and train a video generator with reinforcement learning methods to achieve better outputs.

## D    EXPLANATION EVALUATION PROMPT

```
"""
You will receive a ground-truth explanation and a model-predicted explanation.
Rate the predicted explanation on a scale of 0, 0.5, or 1 based on:
- 0: Completely wrong or no explanation.
- 0.5: Same object mentioned but incorrect reason.
- 1: Both object and reason correctly identified.
Respond with only the score.

Examples:
Ground Truth: 'The clock hands are floating without support.'
Predicted   : 'The clock face is tilted.'
Answer      : 0

Ground Truth: 'The bottle cap is warped at the hinge.'
Predicted   : 'The bottle cap is warped at the hinge.'
Answer      : 1

Ground Truth: 'The lamp bulb flickers due to pixelation.'
Predicted   : 'The lamp bulb is pixelated but stable.'
Answer      : 0.5
"""
```

## E    INFERENCE SETTING AND PROMPT

Models capable of adjusting their frames-per-second (fps) parameter were configured to use an fps value of 2, while the default settings were retained for models lacking this capability. Notably, Video-LLaVa is limited to processing exactly 8 frames using VLMEvalKit (Duan et al., 2024). The Llava-onevision-qwen2-7b-ov-hf model inherently supports only an fps setting of 1 but allows manual configuration of the number of max frames. For our experiments, we set this value to 20. All remaining models utilized the standard configuration of fps=2.

For all models evaluated under zero-shot inference conditions, we maintained the identical baseline prompt as follows:

```
"Decide whether the video is AI-generated or real. "
"If you think it is real, reply 'REAL'. Otherwise, reply 'FAKE' "
"and provide the area coordinates in [x0, y0, x1, y1] format, "
"the starting time of when you think it becomes fake, and an explanation. "
"Encapsulate your answers using <coordinates> </coordinates>, "
"<start_time> </start_time>, and <explanation> </explanation>."
```

## F  QUALITATIVE ANALYSIS

Qualitative analysis can be found in Figure 7.

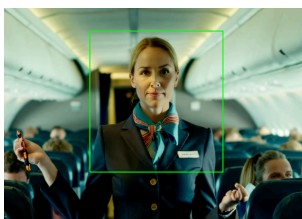 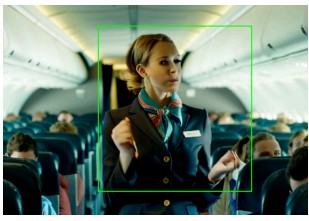 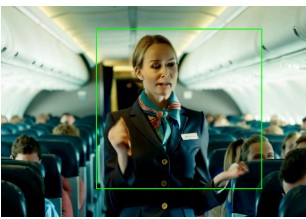

*Ground Truth*: The flight attendant's **hands undergo severe distortion and blurring** while demonstrating the safety instructions, making their shape indistinct and unnatural.

*GPT*: The **faces and expressions of the people**, especially the flight attendant, appear slightly unnatural and lack the subtle imperfections and micro-expressions typical of real humans. The lighting and shadows are overly uniform and do not interact with the environment in a fully realistic way. The details in the background, such as the passengers' faces and the seats, have a slightly artificial, painted look, which is common in AI-generated imagery.

*Ours*: The flight **attendant's hand undergoes severe shape distortion**, appearing unnaturally stretched and altered. This deformation makes the hand look highly unrealistic and inconsistent with normal human anatomy.

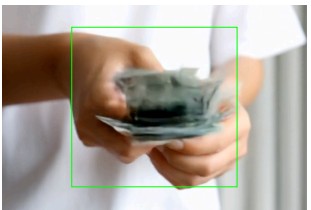 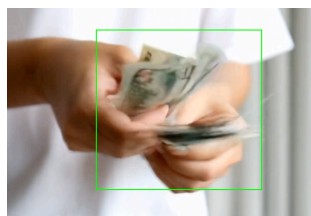 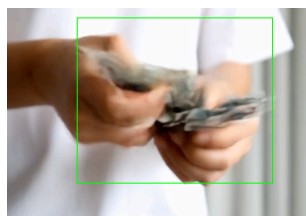

*Ground Truth*: As the person counts the money, **the bills** undergo noticeable deformation, appearing to **warp and bend** unnaturally. Additionally, the edges of the bills **blur** together, making it difficult to distinguish individual notes, creating an inconsistent and unrealistic visual effect.

*GPT*: The **hands and money** have unnatural **blurring** and warping, especially during motion, which is not consistent with typical camera motion blur. The fingers and bills sometimes blend together or lose clear definition, and the edges of the bills appear distorted or melted in some frames. These are common artifacts in AI-generated videos, particularly with current generative models struggling with fine details and fast movement. The overall texture and lighting also appear inconsistent across frames.

*Ours*: The **money in the person's hand** undergoes severe **blurring and shape distortion**, making it difficult to discern the details of the bills. The edges of the money appear to be unclear and warped, creating an unrealistic effect as the money should remain stable.

Figure 7: Qualitative analysis examples that compare ground-truth explanation, GPT generated explanation, and explanation generated by our best 7B reward model based on Video-LLaMa3.

## G  FINETUNING SETUPS

|  | Train Set | Val Set | Test Set |
|---|---|---|---|
| Annotation Count | 3,460 / 2,654 | 434/332 | 440/332 |
| Unique Video Count | 2,654 / 2,654 | 332/332 | 332/332 |

Table 3: **Detailed statistics about the training, val, test data we used.** They are randomly sampled from DEEPTRACEREWARD with ratio being 8:1:1 by unique video. Each cell is reported as fake/real.

**Finetuning Details**   All fine-tuning experiments are conducted on 8 x NVIDIA H100 80GB SXM GPUs. For VideoLLaMA 3 7B base model, one epoch takes around 40 minutes. For Qwen 2.5 VL 7B base model, one epoch takes around 70 minutes.

## H  VAL SET RESULTS

| Base Model | VideoLLaMA 3 - 7B | Qwen 2.5 VL - 7B |
|---|---|---|
| batch size | 1 | 2 |
| fps | 2 | 1 |
| max frame number | 180 | 20 |
| learning rate | $1 \times 10^{-5}$ | $1 \times 10^{-5}$ |
| epoch number | 1 | 1 |
| optimizer | AdamW | AdamW |

Table 4: Hyper-parameter settings for best fine-tuned models, upon the two base models we used.

| Model | Overall ↑ | Acc. ↑ | Fake Acc. ↑ | Real Acc. ↑ | Explanation ↑ | BBox IoU ↑ | BBox Dist. ↓ | Time Dist. ↓ |
|---|---|---|---|---|---|---|---|---|
| | | **Fake *v.s.* Real Classification** | | | **Deepfake Trace Detection** | | | |
| *Baseline Models* | | | | | | | | |
| GPT 5 (Achiam et al., 2023) | 36.3 | 89.7 | 82.7 | 98.8 | 43.4 | 12.0 | 37.8 | 100.0 |
| GPT 4.1 (Achiam et al., 2023) | 37.1 | 91.5 | 86.4 | 98.2 | 34.7 | 22.2 | 25.7 | 100.0 |
| Gemini 2.5 Pro (Team et al., 2023) | 30.0 | 84.3 | 75.7 | 95.8 | 26.7 | 8.9 | 44.7 | 100.0 |
| Gemini 2.5 Flash (Team et al., 2023) | 22.0 | 71.9 | 51.8 | 98.2 | 10.2 | 5.2 | 71.5 | 99.4 |
| LLaVa-One-Vision 7B (Li et al., 2024a) | 11.5 | 45.2 | 38.7 | 53.6 | 0.9 | 0.0 | 97.5 | 100.0 |
| Video-LLaVa 7B (Lin et al., 2023) | 10.8 | 43.3 | 0.0 | 100.0 | 0.0 | 0.0 | 100.0 | 100.0 |
| Phi-4-vision (Abdin et al., 2024) | 8.3 | 33.0 | 0.2 | 75.9 | 0.0 | 0.0 | 99.9 | 100.0 |
| Phi-3.5-Vision (Microsoft, 2024) | 6.4 | 25.5 | 3.9 | 53.6 | 0.0 | 0.0 | 99.1 | 100.0 |
| Qwen 2 VL 7B (Bai et al., 2023) | 15.7 | 59.1 | 30.7 | 96.4 | 3.6 | 0.0 | 86.6 | 100.0 |
| Qwen 2.5 VL 7B (Bai et al., 2025) | 15.3 | 50.5 | 20.7 | 89.5 | 9.1 | 0.6 | 87.9 | 99.2 |
| Qwen 2.5 VL 32B (Bai et al., 2025) | 13.5 | 48.4 | 9.7 | 99.1 | 4.6 | 0.0 | 94.6 | 99.1 |
| Qwen 2.5 VL 72B (Bai et al., 2025) | 17.6 | 50.7 | 17.5 | 94.0 | 8.5 | 0.0 | 90.3 | 88.7 |
| VideoLLaMa3 7B (Zhang et al., 2025) | 9.3 | 36.7 | 2.8 | 81.0 | 0.5 | 0.0 | 100.0 | 100.0 |
| DEEPTRACEREWARD *Models* | | | | | | | | |
| Our (base Qwen 2.5 VL 7B) | 37.8 | 74.2 | 54.4 | 100.0 | 33.3 | 1.5 | 64.5 | 57.7 |
| w/o time | 29.4 | 80.2 | 65.0 | 100.0 | 36.5 | 0.9 | 58.0 | 100.0 |
| w/o explanation | 39.4 | 90.9 | 83.9 | 100.0 | 0.0 | 1.6 | 47.3 | 34.8 |
| w/o time & w/o explanation | 18.5 | 73.1 | 52.5 | 100.0 | 0.0 | 1.0 | 66.4 | 100.0 |
| Our (base VideoLLaMa3 7B) | **68.1** | 97.8 | 96.3 | **99.7** | 66.9 | **32.0** | 16.3 | 24.4 |
| w/o time | 49.6 | 98.2 | 97.7 | 98.8 | **68.1** | **32.0** | 15.0 | 100.0 |
| w/o explanation | 51.7 | 98.4 | 97.9 | 99.1 | 0.0 | 31.6 | 15.1 | **23.1** |
| w/o time & w/o explanation | 32.6 | **98.7** | **98.2** | 99.4 | 0.0 | 31.5 | **14.9** | 100.0 |

Table 5: Val set results on DEEPTRACEREWARD.

