# OpenReview forum: "Learning Human-Perceived Fakeness in AI-Generated Videos via Multimodal LLMs"
_ICLR.cc/2026/Conference — Submitted to ICLR 2026_

### Official Review · Reviewer_89qT · 2025-10-27

**Soundness:** 3
**Presentation:** 3
**Contribution:** 3
**Rating:** 6
**Confidence:** 4

**Summary:**

This work introduces DeeptraceReward, a benchmark and modeling framework designed to capture how humans perceive “fakeness” in AI-generated videos. The authors argue that while video generation quality has improved rapidly, little attention has been given to understanding which specific visual or temporal cues make humans recognize a video as machine-generated. To address this, they collect over 4,300 fine-grained annotations across 3,300 generated videos, each including a natural-language explanation, spatial bounding box, and temporal boundaries of the perceived fake trace. From these annotations, they derive nine major categories of deepfake traces and train multimodal large language models (MLLMs) as reward models to mimic human perception. Their 7B-parameter model surpasses GPT-5 by 34.7% on average across three key dimensions: identifying fake clues, localizing them spatially, and explaining them linguistically. Notably, they uncover a clear hierarchy of task difficulty—binary fake vs. real classification is easiest, followed by explanation, spatial grounding, and finally temporal labeling as the hardest. Overall, the paper provides both a novel dataset and a methodology for modeling human-perceived authenticity in AI-generated videos, advancing research on interpretable and socially aware video generation systems.

**Strengths:**

1. The purpose of DEEPTRACEREWARD is highly in line with the current development needs in video generation field and will be definitely helpful to either improve authenticity of AI generated videos or intelligently identify whether a video is real or AI generated.
2. The paper clearly exposes a limitation of state-of-the-art MLLMs such as GPT-5 and Gemini-2.5-Pro—their inability to spatially and temporally ground fine-grained deepfake traces despite being able to classify videos as fake. The proposed 7B reward model significantly outperforms these baselines, laying a strong foundation for future research in human-aligned video evaluation.
3. The annotation pipeline is well designed and rigorously executed. For instance, the use of LabelBox for structured labeling, meticulous frame-by-frame inspection by annotators, and GPT-4-based standardization of textual explanations all contribute to a reliable and reproducible dataset.
4. The multi-dimensional evaluation metrics effectively demonstrate performance gaps across MLLMs, making the analysis comprehensive and interpretable.

**Weaknesses:**

1. Lack of detailed explanations about the construction process of prompts used to text-to-video generation, which is an indispensable part in such benchmarks related to AI generated videos.
2. Some strong spatial-temporal grounding baselines (e.g., LLaVA-ST, SpaceVLLM) are missing. Given that the defined task combines deepfake recognition and spatial-temporal localization, inclusion of such models would provide a fairer and more informative comparison.
3. The ablation study omits a “w/o bounding box” condition. Since the paper analyzes the effects of time and explanation labels, the contribution of bounding box annotations should also be examined for completeness.

**Questions:**

1. Why do SOTA MLLMs (e.g., GPT-5, Gemini-2.5-Pro) perform well on binary real/fake classification but struggle with fine-grained grounding? Without explicit trace localization, how do these models internally judge fakeness?
2. Why do nearly 40% of the fake videos not have explanation about how they could be identified as fake videos? Does this mean these videos are easy to discern but hard to give corresponding explanations under the defined evaluation framework?
3. Why does training the model w/o explanation do good to metrics like fake accuracy, bounding box distance and time distance? Is there any interpretation for this kind of contrastive effect between “time” and “explanation” labels?
4. When multiple deepfake traces occur at different timestamps in a single video, how did you determine which label and explanation to use for that instance?
5. You mention summarizing nine movement-centric categories after analyzing annotations, yet Figure 3 seems to define tags before annotation. Could you clarify this potential inconsistency?

---

> ### Author Response · Authors · 2025-11-25
>
> Dear Reviewer 89qT,
> We sincerely thank you for your insightful and expert comments and the time you dedicated to reviewing our work. We hope our response and added experiment results could address the concerns. If there’s any place we could provide a more detailed response please let us know!
>
> **Concern [1]**: Video prompt collection details.
>
> **Response [1]**: As in Sec 2.1, we first use the GPT 4.1 model to generate natural and realistic prompts. These prompts are manually filtered and then fed into various text-to-video (T2V) models. Specifically, we require the prompts to involve movement, and cover diverse things including people, animals and objects.
> Some prompts in our datasets are like: *“A riveting martial arts tournament, where fighters demonstrate a blend of discipline and strength, performing precise kicks and powerful blocks in a respectful competition.”, “A scenic horse riding tour set on rolling hills, where riders and stallions unite gracefully over grassy fields and winding rural paths beneath expansive skies”, “A soothing pottery class where participants shape clay and paint, learning crafting techniques in a relaxed and creative environment.”*...etc.
> We will add the detailed prompt construction process and some example prompts into our revision.
>
> **Concern [2]**: Adding spatial-temporal grounding baselines LLAVA-ST and SpaceVLLM.
>
> **Response [2]**: Thank you so much for pointing these two works out. We will definitely add them into our related work section! Indeed, these works are related in terms of localization, while none of them did deepfake video detection training. SpaceVLLM actually did not release any public model checkpoint, or any code at all. LLAVA-ST is a concurrent work with our work so we did not include it as a baseline. Both of these models do not involve any training related to deepfake detection. From our current evaluation on 13 strong baselines, we found that models except GPT or Gemini usually get an overall score ~10 (out of 100) for our task, therefore it won’t be fair comparison to test on LLAVA-ST since it won’t be able to classify fake videos without necessary deepfake training. Despite all these concerns, we still tried to run the LLAVA-ST model, and found that their released model could not be run due to some bugs in their code... We already contacted LLAVA-ST’s authors and we will keep trying to solve the problem and get some results.
>
> **Concern [3]**: Ablation study concerns.
>
> **Response [3]**: We did not include “w/o bounding box” condition because we use this as our “base & necessary” setup – and do ablation study upon this setup, considering if we add temporal or explanation information or not. In this paper, we proposed deepfake trace detection and the bounding box is the most essential input for it. If we do w/o bounding box setting, then we are purely just training our model to do the classification task (which is not the purpose of our work), and the performance scores will be high up till 99+% as according to our previous experiments.
> As for why “w/o explanation” does good to certain metrics – our current intuition is that “numbers” and “language” could be two separate goals during model training. And if we do not require the model to generate textual explanation, maybe it could slightly help on some tasks that require generating numbers, e.g. temporal seconds.

---

> ### Author Response · Authors · 2025-11-25
>
> **Concern [4]**: Why SOTA models can classify but not localize
>
> **Response [4]**: Since we do not have access to the training data of the GPT and Gemini models, we could only make some assumptions here. (We uploaded all model outputs to the supplementary material as a reference!) Based on these models’ outputs, we believe that they may have been trained on some fake v.s. real classification task already on images and videos, but not on those faketrace grounding data (probably could because these data are so hard to annotate). Therefore, we see these performances score on binary classification much higher than groundings. Without localization, they could simply collect some video data e.g. Youtube, and generate some synthetic videos from T2V models, and then train to predict if a video is fake or real, because such data is much easier to get without the need of human annotation.
>
> **Concern [5]**: Annotation Concerns.
>
> **Response [5]**: We have a huge expert annotator group of 11 people focusing on the data annotation. And the human annotation really took us several months to accomplish. During this process, because it is so manual labor demanding, some of the annotators failed or forgot to enter the explanations. But every annotation for sure contains bounding box and temporal information to guarantee spatio-temporal grounding.
> As for the category and tag concerns in Line 252 and Figure 3 – we provided these tags during the annotation process to help annotators as a hint. After the whole annotation was done, we analyzed the explanations only for the categorization analysis. We looked at the explanations, discussed within all annotators, and decided on the categories. Then, we classified the annotations based on their explanations. We will add this clarification in our revision.
>
> **Concern [6]**: Data Concerns.
>
> **Response [6]**: When multiple deepfake traces occur at different timestamps in a single video, we will use both for training. However, for testing data, we choose the videos that only have single deepfake trace annotation to avoid ambiguity. Of course, maybe we could miss some detail and these testing videos maybe have multiple fake traces, but our annotated ones should be the most obvious ones from a human's perspective, and we want the model to be able to at least predict these.
>
> Dear Reviewer 89qT, thank you once again for your insightful and expert comments and the time you dedicated to reviewing our work. If our response addressed your concerns, we really hope you could consider raising your score! If it did not fully answer your questions, please let us know and we will do our best to solve these questions. Thank you.

---

### Official Review · Reviewer_3pS9 · 2025-10-31

**Soundness:** 3
**Presentation:** 3
**Contribution:** 2
**Rating:** 4
**Confidence:** 5

**Summary:**

This paper introduces DEEPTRACEREWARD, a benchmark and set of reward models for human-perceived fakeness in AI-generated videos. Instead of only asking “is this video real or fake,” the authors argue that what really matters is whether humans can point to concrete, spatiotemporal traces that reveal the video as machine-generated (distortions, merging/splitting, sudden blur, odd trajectories, disappearing objects, etc.). They collect AI-generated videos from current SOTA T2V systems (Sora, Kling, Pika, MiniMax, Gen-3, Mochi, curated Sora demos) and annotate fine-grained “deepfake trace” instances with: (i) a natural-language explanation, (ii) a bounding-box over time, and (iii) start/end timestamps. They also pair these with an equal number of real videos from LLaVA-Video-178K for balanced training. They then train multimodal LLM-based reward models (on top of VideoLLaMA3 and Qwen2.5-VL) and show that, on their benchmark, all open/closed baselines (GPT-4.1, GPT-5, Gemini 2.5) are good at real/fake classification but bad at localizing and timing the actual fake trace, while their fine-tuned 7B model gets a big jump in overall score and can localize better.

**Strengths:**

1. This paper is easy to follow.
2. Timely, well-motivated task. Video gen is getting good enough that “why is it fake?” matters; the paper hits that exactly.
3. Fine-grained, human-aligned annotations. Bounding boxes + timestamps + NL reasons across thousands of videos, on current high-end generators, is useful.
4. The motivation is clear and experiments support authors' claim well.

**Weaknesses:**

1. Data-collection pipeline and training setup are not fundamentally new. The contribution is incremental in methodology aspects.
2. Evaluation partly depends on LLM-as-judge for explanations. A human eval slice would make the claim stronger.
3. Generalization to future generators is unclear. See more details in Questions.
4. Incomplete discussion of most recent related work such as [1,2]

Reference
1. How Far are AI-generated Videos from Simulating the 3D Visual World: A Learned 3D Evaluation Approach. 2025
2. BusterX: MLLM-Powered AI-Generated Video Forgery Detection and Explanation. 2025

**Questions:**

1. How robust is the model to unseen generators? If tomorrow’s videos come from, say, a Sora-level model with fewer motion artifacts but more material/lighting artifacts, will the trained reward model still localize well, or will it just say “REAL”? Test on latest models like Sora 2, Kling 2.x etc. would help strengthen the claim.
2. What is the intended release format? Are you releasing raw videos from Sora/Kling, etc., or just annotations + extracted features? This matters for reproducibility/licensing.
3. Happy to see more discussion with latest related work in the related area. The artifact detection capabilities of proposed method is similar to capabilities introduced in [1]. Showing whether the model can detect the physical and geometric artifacts from Sora shown in [1] would help define its capability boundary.

---

> ### Author Response · Authors · 2025-11-25
>
> Dear Reviewer 3pS9,
> We sincerely thank you for your insightful and expert comments and the time you dedicated to reviewing our work. We hope our response and added experiment results could address the concerns. If there’s any place we could provide a more detailed response please let us know!
>
> **Concern [1]**: Novelty of the work & Related works.
>
> **Response [1]**: We want to emphasize that previous works all focus on whole-video reward. E.g. given two videos, choose one that is better than the other one. We are the first work so far to design such inter-video reward, with spatiotemporal awareness, and we believe it is novel in terms of data design till the time we submit our work. As for the training setup – we adopted the standard MLLM training method, since the focus of our paper is more on the reward design. We believe our experiment results could bring much new information that people previously did not know (e.g. GPT5 and Gemini models are really bad at spatial and temporal fakeness localization).
> As for the related works – thanks for pointing them out, and we will definitely cite them in our revisions! They are both very new concurrent works with latest revision last month and this month, and the major differences between our works are that: [1] mainly focuses on 3D world and [2] uses models to generate fake reasons only for the entire video, while we focus on human expert provided inter-video spatial-temporal annotations, which not only include reasons but also include localizations. In other words – our model achieves much better performance than GPT-5, while [2]'s gold annotations come from GPT-kind models.
>
> **Concern [2]**: LLM-as-judge evaluation for explanations concerns.
>
> **Response [2]**: To address this concern, we randomly sampled 10% of the inference outputs from our best-performing model and conducted a human evaluation using the same evaluation prompt employed for the GPT-based assessment, which we included in the appendix. The resulting human-judged score is 73.1%, which is very close to the LLM judged score of 70.6%. We also uploaded all model outputs to the supplementary material for your reference.
>
> **Concern [3]**: Generalization to unseen generators.
>
> **Response [3]**: Definitely this is an important question to current AI research papers. However, we think we did our best – our reward model already achieves the best within a bunch of Sota MLLM models. We believe the development of AI models is always a curvy path. And that our model will be one part of the reason why the future video generation models will become better and better.
> As for testing on the latest released video generation model Kling 2.5 Turbo – since we do not have enough resources to generate many videos from that, we manually generated random 10 videos using free credits. In general, our model is still able to detect the fake videos successfully for all videos and most explanations are acceptable. We think the reason is because most deepfake traces in Kling 2.5 Turbo are similar to what we saw during our annotation on previous Kling models.
> As for Sora2 – unfortunately we do not have access to sora2 at the moment, so we cannot do full evaluations on the same video source prompts. However, we did take a manual look at uploaded Sora2 generated videos from online google search. We believe our model can successfully identify the easy ones, e.g. cartoon ones, or unrealistic ones like flying cats, etc. It might be difficult for our model to detect still videos, like a person with body remaining still but mouth talking. Definitely Sora2 generates motions much more smoothly than Sora, and our model could find it more difficult to detect deepfake traces.
>
> **Concern [4]**: Releasing Format.
>
> **Response [4]**: Yes, we will release the whole set on a public website like Huggingface – including the raw videos, the annotations, and the model predicted outputs (the fine-tuning code for qwen2.5vl and videollama3 are just their official repos.) We are certain this is reproducible and hope our annotated dataset could be useful for future researchers.
>
> Dear Reviewer 3pS9, thank you once again for your insightful and expert comments and the time you dedicated to reviewing our work. If our response addressed your concerns, we really hope you could consider raising your score! If it did not fully answer your questions, please let us know and we will do our best to solve these questions. Thanks.

---

### Official Review · Reviewer_D7KG · 2025-11-01

**Soundness:** 4
**Presentation:** 4
**Contribution:** 4
**Rating:** 8
**Confidence:** 5

**Summary:**

This paper introduces DEEPTRACEREWARD, a dataset designed to identify and analyze human-perceived flaws/anomalies in AI-generated videos. The authors created a dataset of over 4,300 annotations on 3,300 videos, where each inconsistency, or "deepfake trace," is documented with a natural language explanation, a bounding box, and precise start and end times. Using this data, they trained an MLLM as a baseline to act as a reward model that mimics human judgment in detecting these artifacts. This model significantly outperformed state-of-the-art systems like GPT-5 and Gemini 2.5 Pro, and the research revealed a clear hierarchy of difficulty: explaining a flaw is easier than grounding it in visual cues. And localizing where the fake part begins seems to be the hardest problem of all.

**Strengths:**

- The paper provides a much needed and timely expert annotated video dataset for AI-Generated video detection and reasoning, as it is largely missing in the literature. Most efforts have gone into image reasoning datasets, which has become largely insufficient in today's day and age.
- The scale of the dataset is also sufficient for future models to finetune on.

**Weaknesses:**

- The paper mentions that low-quality generated videos were filtered out. However, annotations for these low-quality videos are especially essential, because more often than not, the DeepFakes appearing in social media are low-quality. If a model is only trained on high-quality annotated videos, in practical deployment applications, this will result in a significant bias.
- All "real" videos were sourced from a single dataset (LLaVA-Video-178K). This raises a siginificant concern that the model (or any model trained on this data) just might be distinguishing between the sources of the data as a "shortcut" rather than actually grounding it's detection/reasoning on visual cues.  To prevent issues from domain gap, real videos could have been sourced just as easily from other publicly available licensed sources like FF++, CelebDF, etc.
- Since the main contribution of the paper is focused on providing expert annotations for the videos, the missing performance metrics like BLEU, ROUGE-L, CIDEr, etc. is a critical limitation. Without these NLG metrics it is hard to verify how well the trained model is actually reasoning.
- The dataset is also missing face-manipulated videos and is focused on T2V/I2V generators. Face-manipulated videos are nuanced since a large (spatial) chunk of the video is real and the only anomaly is in the face region. It will be extremely helpful to have some videos from this traditional DeepFake literature (for example videos from FF++, CelebDF, etc.).

**Questions:**

- How was consensus reached among annotators when there were disagreements on labeling an artifact? What was the protocol for resolving subjective ambiguities?
- The results show that temporal localization (predicting the start time of a fake trace) is the most difficult task for all models. What is the authors' hypothesis for why this is the case? Is it a limitation of current model architectures/training paradigms or an issue of ambiguous annotations? For example, a frame-by-frame model should be able to easily tell where the fake frame begins, as long as we assume it is a reasonably good (oracle) Deepfake detector.
- The ablation study indicates that removing supervision for explanations or temporal data can sometimes improve the model's performance on other metrics. Why might training on multiple tasks (e.g., localization and explanation) simultaneously cause this interference?
- A recent paper: PhysicsIQ [1] showed that Sora has the high visual realism and very low physical realism (i.e., following principles of gravity, fluid mechanics, etc.) in it's generated videos. Are the performance of the tested models on the annotated Sora videos available? It will be extremely valuable to see how much of a difference the provided annotations make on detecting or reasoning on Sora videos.
- How do the authors plan to keep the benchmark relevant as video generation technology rapidly evolves and produces new kinds of artifacts? Will it be updated periodically?
- The video data along with the explanations will be made publicly available for at least academic use, correct? And what is the expected timeline for the data release? This is crucial for the community.

[1] Motamed, Saman, et al. "Do generative video models understand physical principles?." arXiv preprint arXiv:2501.09038 (2025).

---

> ### Author Response · Authors · 2025-11-25
>
> Dear Reviewer D7KG,
> We sincerely thank you for your insightful and expert comments and the time you dedicated to reviewing our work. We hope our response and added experiment results could address the concerns. If there’s any place we could provide a more detailed response please let us know!
>
> **Concern [1]**: Data Bias Concerns.
>
> **Response [1]**: During our video collection, we filtered out very low-quality generated videos. For those, human annotators found it too hard because “the whole video just seems so fake”, e.g. the entire video could be too vague or blurring, or be shaking, or lack discernible content…, etc. Annotating such videos would require drawing a bounding box over nearly every frame region, offering little meaningful supervision. Our current dataset therefore focuses on videos where at least some localized deepfake traces are visually identifiable, which still reflects the majority of “low-quality fakes” commonly seen on social media. We acknowledge, however, that excluding the most degraded samples may introduce bias, and agree that expanding annotation to a broader range of difficult cases is an important direction for future work.
>
> As for the real videos that we used during training – we sampled them from LLaVA-Video-178K because it has been one of the largest video datasets that combines multiple previous video dataset sources covering diverse topics. We totally agree that adding real videos from other publicly available licensed sources like FF++, CelebDF, etc. will definitely help prevent issues like domain gaps further. However, with our comprehensive evaluation with 7 metrics, we believe if the model relies on a shortcut we will be able to tell from our metric results. Currently the explanation, grounding, and accuracy scores are somewhat consistent, suggesting that our current model doesn’t have severe shortcuts.
>
> **Concern [2]**: Metric Concerns.
>
> **Response [2]**: We agree that adding BLEU/ROUGE-L/CIDEr will make the results more complete, but we have concerns that they are not suitable for deepfake-trace explanations. These explanations are usually short and semantically precise, and n-gram overlap could correlate poorly with human judgments (e.g. we observed many correct explanations with near-zero BLEU/ROUGE because they’re paraphrases). Thus, following recent LLMs evaluation suites like opencompass and VLMEvalKit, we use an LLM-as-judge method to provide a far more reliable measure of whether the model actually identifies the visual artifact being explained.
>
> **Concern [3]**: Face-manipulated Videos Missing Concerns.
>
> **Response [3]**: We agree that incorporating face-manipulated videos would make the benchmark even more comprehensive. Our current work focuses on modeling human-identified fake traces in a broader, general sense rather than targeting a specific manipulation type. Extending the dataset to include fine-grained face-manipulation categories is indeed valuable, and we view this as an important direction for future work.
>
> **Concern [4]**: Annotator Consensus Concerns.
>
> **Response [4]**: As we mentioned in Line 207-215, the annotation process poses several challenges. A primary issue was subjective ambiguity – annotators occasionally struggled to distinguish between closely related artifact types (e.g., object merging vs. object disappearance). To mitigate this problem, we adopt a consensus-based workflow in which annotators collaboratively look at the same ambiguous case and agree with the majority-vote results. Each annotated video is then reviewed by one to two additional annotators to ensure cross-validation and minimize bias. We added additional experiments to calculate inter-annotator agreement. Due to the huge human labor requirement, we randomly sampled 50 examples from the dataset that two of our annotators have not seen before and asked them to annotate them to calculate Cohen's Kappa for fake type categorization and average BBox IoU localization. For categorization, we reach a kappa of 0.776 in this 9-class classification task, showing substantial agreement. For the task of bounding box annotation, we calculate the average IoU between the two annotators and result in 0.624. We also ran a simple experiment for a random baseline on this IoU metric where we uniformly sample two sets of coordinates for two bounding boxes for 100 runs in a 640*480 settings and result in an average IoU of 0.038, showing the difficulty of this annotation and inference task. We will add and discuss the above information in the revisions.

---

> ### Author Response · Authors · 2025-11-25
>
> **Concern [5]**: Why temporal localization is most difficult.
>
> **Response [5]**: We have several assumptions. First, natural temporal localization data could be most sparse in natural video data, compared to natural language or object localization data that is already quite common in image datasets. Temporal localization data is video specific, and cannot be shared with the large amount of image data. Second, we are the first to propose to annotate deepfake trace temporal localization data, and the annotation indeed took a lot of time – several months for a group of 11 expert annotators. Collecting such data is just hard. Third, the current vision encoder design in MLLM models is problematic. Especially when the video is way too long, then the vision encoder can only set a small enough FPS value and take limited frames. This design is not good for the MLLM model to get a sense of temporal information.
>
> **Concern [6]**: Multi-task interference
>
> **Response [6]**: This phenomenon is expected in multitask learning. Our current intuition is that “numbers for localization” and “language for explanation” could be two separate goals during model training. And if we do not require the model to generate textual explanation, maybe it could slightly help on some tasks that require generating numbers, e.g. temporal seconds.
>
> **Concern [7]**: Result Availability
>
> **Response [7]**: Yes! We just uploaded all model outputs to this submission’s supplementary material! As for Sora results – we need to emphasize that we did not collect many Sora videos, because the model was not available to the public and in total we only collected 38 videos manually from their website, and 3 were included in test data with 2 included in val data. We attach the evaluation for sora videos as below. Performance of our model on the sora videos is indeed lower than that on all videos, suggesting that sora is indeed a good model and hard to detect fake traces.
> | Our Model              | Average of Sora | Average of all test data |
> |--------------------|-----------|------------------|
> | classification_score↑ | 100       | 99.4             |
> | fake_accu↑            | 100       | 99.6             |
> | explanation_score↑    | 40        | 70.6             |
> | bbox_iou↑             | 8.9       | 32.6             |
> | bbox_distance↓        | 34.2      | 13.6             |
> | time_distance↓        | 38.6      | 21.9             |
>
> **Concern [8]**: Benchmark Updates and Availability
>
> **Response [8]**: We will try our best to update the benchmark periodically, especially when popular next generation of video-generation models becomes accessible to the public.
> As for availability – Yes, all data, code, model will be publicly available for academic usages. We will release it anytime since everything is already ready.
>
> Dear Reviewer D7KG, thank you once again for your insightful and expert comments and the time you dedicated to reviewing our work. If it did not fully answer your questions, please let us know and we will do our best to solve these questions. Thanks.

---

### Official Review · Reviewer_9QQJ · 2025-11-02

**Soundness:** 3
**Presentation:** 3
**Contribution:** 2
**Rating:** 4
**Confidence:** 4

**Summary:**

This paper addresses a critical and timely challenge in the evaluation of AI-generated video: moving beyond simple binary classification (real vs. fake) to understand why, where, and when humans perceive artifacts in synthetic content.

I observe following contributions:

- A novel benchmark designed to capture "human-perceived deepfake traces." It comprises 3.3K high-quality generated videos sourced from seven SoTA models (including Sora, Kling, Pika) and 3.3K real videos. It features 4.3K fine-grained expert annotations, each including a natural language explanation, spatial localization (bounding boxes), and temporal localization (timestamps).

- The authors analyze these annotations to develop a taxonomy of nine major categories of movement-centric artifacts (e.g., Object Distortion, Object Merging, Sudden Blurring).

- An extensive evaluation of 13 MLLMs. The results show that while SoTA models may achieve high accuracy in binary classification, they perform poorly (below 37% overall) on the fine-grained grounding and explanation tasks.

- The authors demonstrate the dataset's utility by fine-tuning a 7B model (based on VideoLLaMA 3), which achieves 70.2% overall performance, significantly outperforming the baselines.

- The paper identifies a consistent difficulty gradient: binary classification is easiest, followed by explanation, spatial grounding, and finally, temporal localization (hardest).

**Strengths:**

As video generation models approach photorealism, the need for interpretable, localized, and explainable detection methods becomes paramount. This motivation makes the work relevant. This work provides a framework for understanding the specific failure modes of current generators.

- Novel and Important Task Formulation: The shift towards interpretable, grounded reasoning for deepfake detection is essential for trustworthy AI and for improving generation models via human feedback. This approach significantly advances the field beyond existing benchmarks like VBench (holistic quality) or DeMamba (binary classification).
- Valuable Dataset Contribution: DEEPTRACEREWARD is a meticulously constructed and highly relevant resource. The effort to collect expert annotations with multimodal grounding (text, space, time) across diverse SOTA generators is commendable. The focus on high-quality, motion-rich videos ensures the benchmark addresses the subtle, temporally dependent artifacts characteristic of cutting-edge models.
- *Insightful Taxonomy of Failures:* The categorization of the nine major artifact types (Sec 2.3, Figs 1 and 6) provides excellent insight into the current limitations of generative models, particularly regarding temporal consistency, physics, and object permanence (e.g., splitting, merging, disappearance).
- Good Empirical Validation and Analysis: The extensive benchmarking clearly highlights the deficiencies of current MLLMs for this task. The significant performance leap of the fine-tuned 7B model (from <37% to 70.2%) strongly validates the quality of the dataset as a supervisory signal. The analysis of the difficulty gradient (L424-430) is an important empirical observation, pinpointing temporal reasoning as a major bottleneck.

**Weaknesses:**

- Lack of reproducibility: The repeated reference to "GPT 5" and "GPT 4.1" (e.g., Abstract, Table 2, Section 3.1). The citation provided, (Achiam et al., 2023), refers to the original GPT-4 technical report. These names do not align with standard public nomenclature (e.g., GPT-4o, GPT-4 Turbo).

This ambiguity severely undermines the credibility and reproducibility of the baseline comparisons.
The central claim that the authors' 7B model "outperforms GPT-5 by 34.7%" (L025-026) cannot be verified. The authors must specify the exact model identifiers (e.g., API endpoints or version numbers) used in the experiments or even then these baselines are blackboxed (what if the authors were part of an A/B test?)

- Inflated Metric: The Overall score (L380) is an unweighted average of four metrics, including binary accuracy. As the paper's model achieves near-perfect binary classification (99.4%), a task acknowledged as easy (L27), this heavily inflates the overall 70.2% result. This masks the much lower performance on the novel grounding tasks (e.g., BBox IoU of 32.6%) and presents a misleading picture of the model's capabilities.

- Inadequate Temporal Localization: The "Time Distance" metric (L375) only measures the error in the starting timestamp, ignoring the duration of the artifact. A Temporal IoU metric, which considers the overall temporal overlap, is standard in video analysis and required here.

- The paper is heavily framed around providing a "Reward" signal for video generation (Title, Abstract and throughout the paper). However, the experiments focus entirely on video understanding/detection. While the trained model could serve as a reward model (e.g., for RLHF), this utility is hypothesized (L711) but not empirically demonstrated in the paper.

**Questions:**

- Why does the paper use GPT-4 to post-process and standardize the explanation text (L:213)? Does this not introduce LLM biases / randomness and undermines the premise that the paper is capturing human perception by effectively contaminating the ground truth?

- To add to above question: Please explain how does the Explanation score relying solely on LLM judgment (L357) substitutes for the focus on human perception (As per the research question in intro). Given the contamination of the ground truth by GPT-4, this metric may be evaluating alignment with LLM-standardized text rather than genuine human explanations.

- Who were the LabelBox experts?

- What was the inter-annotator agreement? It can be measured via Cohen's Kappa for categorization and average IoU for localization.

**Details Of Ethics Concerns:**

Annotators are unknown and no information about them or how/how-much they are compensated is provided.

---

> ### Author Response · Authors · 2025-11-25
>
> Dear Reviewer 9QQJ,
> We sincerely thank you for your insightful and expert comments and the time you dedicated to reviewing our work. We hope our response and added experiment results could address the concerns. If there’s any place we could provide a more detailed response please let us know!
>
> **Concern [1]**: About GPT Baseline Reproducibility: unclear setting with the GPT baselines, especially the specific API endpoint or version number, and hard to verify if our 7B model really out-performs GPT-5.
>
> **Response [1]**: For the GPT-5 and GPT-4.1 models that we tested on, we used the endpoints: gpt-5-2025-08-07 and gpt-4.1-2025-04-14. Just for clarification, they are exactly the standard public nomenclature on the official website (https://platform.openai.com/docs/models/gpt-5, https://platform.openai.com/docs/models/gpt-4.1), and both dated-versions are the only versions under the specific model name. While these models are from OpenAI and we do not have control over them, there seems to be no new releases under these two specific versions (newer ones are under the name gpt-5.1). Additionally, we upload the previous evaluation outputs into the supplementary material as a reference.
>
> **Concern [2]**: Evaluations concerns: overall score metric might bias towards the easy task of classification and ignoring the hard task of BBox_IoU.
>
> **Response [2]**: For each of the seven evaluation metrics in Sec 3.1, we scaled them to 0-100, while some of them are higher the better (e.g. accuracy), and some of them are lower the better (temporal distance). Then, we did the unweighted average on Line 380: overall = avg[Accuracy, Explanation_score, BBox_IoU, (100 − Time_distance)].
> To address this concern, we provide new experiment results under new metrics that do not consider accuracy: (A) avg[Explanation_score, BBox_IoU, (100 − Time_distance)], (B) avg[Explanation_score, BBox_IoU, (100 − BBox_distance), (100 − Time_distance)], comparing to (C) the original overall score.
> | Model                     | A: avg[Explanation_score, BBox_IoU, (100 − Time_distance)]      | B: avg[Explanation_score, BBox_IoU, (100 − BBox_distance), (100 − Time_distance)]      | C (Original Overall): avg[Accuracy, Explanation_score, BBox_IoU, (100 − Time_distance)] |
> |---------------------------|--------|--------|-----------------------|
> | GPT-5                     | 17.1   | 28.58  | 35.5                  |
> | GPT-4.1                   | 17.7   | 32.35  | 36.5                  |
> | Gemini 2.5 Pro            | 11.9   | 22.75  | 30                    |
> | Gemini 2.5 Flash          | 5.87   | 12.23  | 23                    |
> | Qwen2.5 VL 7B             | 3.73   | 5.83   | 15.7                  |
> | Qwen2.5 VL 32B            | 2.23   | 2.8    | 13.5                  |
> | Qwen2.5 VL 72B            | 6.43   | 7.15   | 17.3                  |
> | VideoLLaMA-3 7B           | 0.6    | 0.45   | 10                    |
> | Our Qwen 2.5 VL 7B (base) | 26.27  | 28.95  | 38.4                  |
> | Our VideoLLaMA-3 7B (base)| 60.43  | 66.93  | 70.2                  |
> | Diff (Our - GPT-5)        | 43.33  | 38.35  | 34.7                  |
>
> We can see that including accuracy should generally improve model scores (C > A or B), and that our model achieves an even larger advantage over GPT-5 (A: 43, B: 38 > C: 34.7) if we do not consider accuracy scores in the overall scores.
>
> **Concern [3]**: About temporal localization concerns.
>
> **Response [3]**: We agree that temporal IoU is commonly used for videos. However, in the context of deepfake traces, the end boundary of an artifact is inherently ambiguous and produces extremely low inter-annotator agreement during dataset construction. When we annotated the dataset, we found that deepfake inconsistencies often fade out gradually or transition into normal frames without a clear stopping point or boundary, making the duration highly subjective and noisy. In contrast, annotators consistently agreed on the start of the artifact—the moment humans first perceive something as “fake.” Therefore, evaluating only the start timestamp provides a far more stable and perceptually meaningful metric than IoU under our specific task setup and annotated dataset. We will clarify this in the revision.

---

> ### Author Response · Authors · 2025-11-25
>
> **Concern [4]**: RLHF experiment concerns.
>
> **Response [4]**: Our goal in this paper is to provide a good reward signal design for video generation, that can really align with us real human beings. For this purpose, we (entire author team) annotated the whole dataset for several months, and further did the reward modeling experiments on a large set of models. We want to emphasize that this is the first time we set up such a comprehensive inner-video spatiotemporal-aware reward, compared to previous methods. Also, previous popular reward bench papers such as [RewardBench: Evaluating Reward Models for Language Modeling] also focused on the reward bench construction, and did not include the RL experiments. Previous reward modeling papers such as [Generative Reward Models] only did reward modeling, and did not construct reward benchmarks or conduct RL experiments. We definitely agree that adding the RL experiment would increase the value of this paper, but we believe it’s beyond the focus of our paper, and we will leave it to future studies.
>
> **Concern [5]**: Why we polish human explanations with GPT model & Explanation Score Eval.
>
> **Response [5]**: This is one step adopted by our annotators during the annotation process, not a process that we apply after the annotation is done. We do this mainly because humans can make grammar mistakes, and also that sometimes we write too short sentences. This does not introduce LLM bias because when we annotate, we look at the GPT polishing and decide whether we use it or not. Every annotation is final-checked by humans before being added to the dataset.
> Therefore, without the contamination of the ground truth by GPT model, we believe it would not affect the LLM-judge on the explanation score. We do provide our evaluation prompt as in Appendix D on Line 718-737 for reference. We manually went through the LLM judge results during evaluation to make sure the LLM judge was good quality.
>
> **Concern [6]**: Who are the LabelBox experts and what is the inter-annotator agreement.
>
> **Response [6]**: 11 of the paper’s authors are domain expert annotators and we are the only annotators. We used the platform LabelBox to annotate the videos. As for the inter-annotator agreement, due to the huge human labor requirement, we randomly sampled 50 examples from the dataset that two of our annotators have not seen before and ask them to annotate them to calculate the Cohen’s Kappa for fake type categorization and average BBox IoU localization. For categorization, we reach a kappa of 0.776 in this 9-class classification task, showing substantial agreement. For the task of bounding box annotation, we calculate the average IoU between the two annotators and result in 0.624. We also ran a simple experiment for a random baseline on this IoU metric where we uniformly sample two sets of coordinates for two bounding boxes for 100 runs in a 640*480 settings and result in an average IoU of 0.038, showing the difficulty of this annotation and inference task. We will add and discuss the above information in the revisions.
>
> Dear Reviewer 9QQJ, thank you once again for your insightful and expert comments and the time you dedicated to reviewing our work. If our response addressed your concerns, we really hope you could consider raising your score! If it did not fully answer your questions, please let us know and we will do our best to solve these questions. Thanks.

---

### Meta-Review · Area_Chair_bQ3a · 2026-01-02

**Summary:**

The paper provides a benchmark designed to capture "human-perceived deepfake traces", which addresses a critical challenge in the evaluation of AI-generated video: moving beyond simple binary classification (real vs. fake) to understand why, where, and when humans perceive artifacts in synthetic content.  Although very interesting, the dataset has several limitations and need further improvements.

**Reviewer Concerns:**

1. Data Bias Concerns: low-quality generated videos were filtered out; all "real" videos were sourced from a single dataset (LLaVA-Video-178K); missing face-manipulated videos.
2. Overall score metric might bias towards the easy task of classification and ignoring the hard task.
3. Inadequate Temporal Localization: without temporal duration of the artifact.

**Reviewer Scores:**

9QQJ would keep the score.
D7KG would keep the score
3pS9 would increase the score, but not much.
89qT would keep the score

---

### Decision · Program_Chairs · 2026-01-26

Reject